# Optical Coherence Tomography Angiography in Diabetes and Diabetic Retinopathy

**DOI:** 10.3390/jcm9061723

**Published:** 2020-06-03

**Authors:** Jacqueline Chua, Ralene Sim, Bingyao Tan, Damon Wong, Xinwen Yao, Xinyu Liu, Daniel S. W. Ting, Doreen Schmidl, Marcus Ang, Gerhard Garhöfer, Leopold Schmetterer

**Affiliations:** 1Singapore Eye Research Institute, Singapore National Eye Centre, Singapore 169856, Singapore; jacqueline.chua.y.m@seri.com.sg (J.C.); ralene_sim1995@hotmail.com (R.S.); bingyao.tan@ntu.edu.sg (B.T.); damon.wong@ntu.edu.sg (D.W.); xinwen.yao@ntu.edu.sg (X.Y.); liu.xinyu@seri.com.sg (X.L.); daniel.ting.s.w@singhealth.com.sg (D.S.W.T.); Marcus.Ang@Singhealth.com.sg (M.A.); 2Academic Clinical Program, Duke-NUS Medical School, Singapore 169857, Singapore; 3SERI-NTU Advanced Ocular Engineering (STANCE), Singapore 639798, Singapore; 4Institute of Health Technologies, Nanyang Technological University, Singapore 639798, Singapore; 5Department of Clinical Pharmacology, Medical University of Vienna, 1090 Vienna, Austria; doreen.schmidl@meduniwien.ac.at (D.S.); gerhard.garhoefer@meduniwien.ac.at (G.G.); 6Center for Medical Physics and Biomedical Engineering, Medical University of Vienna, 1090 Vienna, Austria; 7Institute of Molecular and Clinical Ophthalmology, CH-4031 Basel, Switzerland

**Keywords:** optical coherence angiography, diabetes, diabetic retinopathy, retinal perfusion density, choriocapillaris flow voids

## Abstract

Diabetic retinopathy (DR) is a common complication of diabetes mellitus that disrupts the retinal microvasculature and is a leading cause of vision loss globally. Recently, optical coherence tomography angiography (OCTA) has been developed to image the retinal microvasculature, by generating 3-dimensional images based on the motion contrast of circulating blood cells. OCTA offers numerous benefits over traditional fluorescein angiography in visualizing the retinal vasculature in that it is non-invasive and safer; while its depth-resolved ability makes it possible to visualize the finer capillaries of the retinal capillary plexuses and choriocapillaris. High-quality OCTA images have also enabled the visualization of features associated with DR, including microaneurysms and neovascularization and the quantification of alterations in retinal capillary and choriocapillaris, thereby suggesting a promising role for OCTA as an objective technology for accurate DR classification. Of interest is the potential of OCTA to examine the effect of DR on individual retinal layers, and to detect DR even before it is clinically detectable on fundus examination. We will focus the review on the clinical applicability of OCTA derived quantitative metrics that appear to be clinically relevant to the diagnosis, classification, and management of patients with diabetes or DR. Future studies with longitudinal design of multiethnic multicenter populations, as well as the inclusion of pertinent systemic information that may affect vascular changes, will improve our understanding on the benefit of OCTA biomarkers in the detection and progression of DR.

## 1. Introduction

Diabetic retinopathy (DR) is an important rising cause of blindness globally [1]. The initial and follow-up evaluation of patients with diabetes has been based on dilated ophthalmoscopy, fundus color photography, fluorescein angiography and optical coherence tomography (OCT). Emerging imaging technologies, such as OCT angiography (OCTA), may further improve the diagnosis and management of the disease and aid us with a better understanding of DR. OCTA is a relatively fast, non-invasive technique that can generate high-resolution images of the retinal microvasculature at distinct depths, allowing an improved delineation of vascular features seen in DR. In this review, we will focus on the clinical applicability of OCTA-derived quantitative metrics that appear to be relevant to the clinical diagnosis, grading, and management of diabetes or DR.

## 2. Epidemiology of Diabetes Mellitus

Diabetes mellitus, more simply referred to as diabetes, is a group of chronic, progressive diseases characterized by high levels of blood glucose. Around the world, an estimated 463 million adults (9.3% of all adults aged 20–79 years) are living with diabetes in 2019, compared to 151 million in 2000 [2]. Importantly, the number of people with diabetes is expected to reach 700 million (10.9%) in 2045 [2].

When diabetes is poorly controlled, it can damage the vasculature, often separated into either damage to the large blood vessels of the body (macrovascular complications i.e., coronary artery disease, peripheral artery disease, and stroke), or small vascular damage (microvascular complications i.e., retinopathy, nephropathy, and neuropathy), which are the major causes of morbidity and mortality in diabetes [3].

## 3. Epidemiology of Diabetic Retinopathy

Diabetic retinopathy is recognized as an important cause of blindness [4], that places a considerable strain on the quality of life [5]. Globally, in 2010, DR is ranked as the fifth most common cause of preventable vision loss, accounting for 1.9% of visual impairment and 2.6% of blindness [4]. Notably, the number of people with visual impairment due to DR worldwide has been steadily increasing for the past two decades [1]. Given that vision loss from DR is preventable, the World Health Organization (WHO) has placed DR on its priority list of eye conditions [6].

Approximately a third of the diabetes population has some clinically visible DR, while a further one-third of those will develop the vision-threatening state of the disease, characterized by proliferative DR or diabetic macular edema [7]. However, considerable heterogeneity exists in the prevalence of DR across the world and within countries [8].

## 4. Pathophysiology and Stages of Diabetic Retinopathy

The pathophysiology of DR is complex and multifactorial [9,10]. Chronic hyperglycemia impacts at least five interrelated pathways: polyol pathway activation; production of advanced glycation endproducts (AGEs); protein kinase C (PKC) activation; hexosamine pathway activation; and poly (ADP-ribose) polymerase upregulation. This, in turn, leads to oxidative stresses, resulting in mitochondrial dysfunction, deregulation of proinflammatory mediators and importantly, hypoxia. These effects cause apoptosis of vascular and neuronal cells and the upregulation of vascular endothelial growth factor (VEGF) expression, eventually leading to neurovascular dysregulation, and hyperpermeable blood vessels and/or neovascularization and eventually give rise to the clinical picture of DR. Notably, the generation of reactive oxygen species (ROS) and oxidative stress further worsens metabolic dysfunction. Moreover, the renin angiotensin aldosterone system is another driver of neurovascular dysfunction. Even though both vascular and neural abnormalities occur in DR, the clinical manifestation of DR is mostly characterized by a series of vascular changes.

From a clinical perspective, DR can be broadly classified into two stages: non-proliferative and proliferative. In the early non-proliferative stage, the loss of pericytes and endothelial dysfunction results in the weakening in the wall of retinal capillaries, which are seen as microaneurysms and develop into exudative changes (leakage of lipoproteins (hard exudates) and blood (blot hemorrhages)) from the damaged vasculature [11,12]. Localized capillary nonperfusion results in regions of ischemia (infarcts of the nerve-fiber layer represented as cotton-wool spots). As the ischemia worsens, DR progresses into the proliferative stage, where new abnormal blood vessels (neovascularization) can develop in the retina, optic disc, and possibly the iris [13]. The formation of diabetic macular edema, a breakdown of the inner endothelial blood-retinal barrier, is characterized by increased vascular permeability, thickening, and the deposition of hard exudates within the macula, and can occur at any stage of DR [14] (Figure 1).

## 5. Diagnosis and Stages of Diabetic Retinopathy

The key to reduce diabetes-related blindness is early detection, considering the availability of numerous treatment options, i.e., laser, anti-VEGF agents, and steroids [15]. The diagnosis and staging of DR have been based on a dilated stereoscopic examination of the posterior pole and color fundus photography. The gold standard for classifying DR has been the Early Treatment of Diabetic Retinopathy Study (ETDRS) classification system, which relies on comparing the numbers of various vascular features with standard reference colored fundus photographs [16]. The ETDRS classification system, though valid and reproducible, remains laborious to execute, which limits its clinical use. Instead, the International Clinical Diabetic Retinopathy and Diabetic Macular Edema Severity scale was developed [17]. The new simplified grading system classified retinopathy severity into five scales and diabetic macular edema into two tiers. Importantly, this grading system was shown to correlate well with the risk of disease progression [17].

## 6. Ocular Vascular Changes in Diabetes

Retinal vascular changes are well-documented in diabetes and DR. Damage to the blood-retinal barrier breakdown represents the first consequence of DR [18,19,20]. An impairment of blood-retinal barrier breakdown integrity leads to the accumulation of fluid in the intraretinal layers of the macula, known as diabetic macular oedema. The mechanism of the blood-retinal barrier breakdown is multifactorial and has been summarized by Bhagat et al. [21] Briefly, it is secondary to alterations in the tight junctions, pericyte loss, endothelial cell loss, retinal vessel leukostasis, up-regulation of vesicular transport, increased permeability of the surface membranes of retinal vascular endothelial and retinal pigment epithelium cells, activation of the AGE receptor, downregulation of glial-cell derived neurotropic factor (GDNF), retinal vessel dilation, and vitreoretinal traction [21].

The breakdown of the blood retinal barrier is associated with vascular basement membrane thickening due to enhanced production of vascular basement-membrane components [22]. This leads to altered communication between endothelial cells and pericytes [23] and subsequent pericyte loss and endothelial cell damage [24]. During the course of the disease, endothelial cells also die, leading to acellular vessels and subsequent vasodegeneration, which is clinically seen as vascular dropout [25].

While the initial triggers are directly a result of the metabolic dysregulation inherent to hyperglycemia, the subsequent progression of DR is related to the neuronal exposure to toxins crossing the damaged blood-retinal barrier and ischemic damage as a result of impaired blood supply due to the vascular occlusion [9]. Retinal ischemia leads to neuronal damage (and eventual death) and the retinal non-perfusion likewise triggers the release of growth factors including VEGF, which promote angiogenesis—proliferation of new vessels (neovascularization), the defining characteristic of proliferative DR.

Given that pericytes and endothelial cells are key players in regulating vascular tone, it does not come as a surprise that diabetes and DR are associated with vascular dysregulation that has been confirmed in a wide variety of clinical and experimental studies. Neurovascular coupling or functional hyperemia, which describes the adaptation of local perfusion to local changes in neuronal activity [26,27,28] is altered in the diabetic retina [29,30,31,32]. A recent study utilized OCTA to assess the vascular reactivity of retinal capillaries in DR eyes and found a reduced vasoconstriction in response to breathing 100% oxygen [33]. An impaired vascular response to hyperoxia is consistent with the impaired vascular autoregulation in DR. This is at least partially a consequence of hyperglycemia [34,35], and associated with endothelial dysfunction [36,37]. In addition, diabetes is characterized by abnormal retinal and optic nerve head vascular autoregulation [38,39,40,41,42], which refers to the adaptation of vascular tone in response to changes in ocular perfusion pressure [43]. It is less well-established how retinal blood flow under resting conditions behaves in diabetes and DR. Evidence for both increased [44,45,46,47,48,49] and decreased retinal blood flow [50,51,52,53,54] have been reported in diabetes and early DR. A recent study suggested that the increased retinal blood flow is a consequence of early retinal hypoxia [49], which is associated with retinal vasodilatation [55,56].

While the signs of DR are often documented as vascular changes, studies have also shown early neural damage in diabetes [9]. Some investigators have suggested that retinal neurodegeneration may be present before the development of clinically detectable microvascular damage, particularly in the reduction of the peripapillary retinal nerve fiber layer thickness [57,58,59,60,61,62]. Such conclusions are, however, difficult to draw, because the results are largely dependent on the sensitivity of the technology that has been used to detect the changes. Indeed, other studies have found opposite results, indicating that vascular changes precede neural damages [63].

## 7. Fluorescein Angiography

Imaging modalities, such as fluorescein angiography, B-scan ultrasonography, and OCT, can be incorporated with exam techniques, depending on the manifestation of DR [64]. Fluorescein angiography uses sodium fluorescein, a fluorescent mineral-based dye to assess vascular integrity, and leakage, and is considered as the current ‘‘gold standard” for evaluating the vasculature in DR. Fluorescein angiography can show microaneurysms (seen as punctate areas of hyperfluorescence), areas of nonperfusion (seen as sparse areas of hypofluorescence surrounded by large retinal vessels), and abnormal blood vessels, such as intraretinal microvascular abnormalities or retinal neovascularization. Apart from assessing DR severity, fluorescein angiography can also indicate the presence of diabetic macular edema, because fluorescein can leak out of incompetent vessels when there is a breakdown of the blood-retinal barrier. In spite of its clinical utility in DR, fluorescein angiography has a number of disadvantages, such as its invasive nature, requiring intravenous dye injection that is associated with serious adverse effects, ranging from nausea to rare but severe anaphylactic shock [65,66].

## 8. Optical Coherence Tomography

OCT is akin to performing an optical biopsy of the retina, where different layers of the retina can be visualized [67]. OCT works by illuminating the retina and then measuring the flying time it takes for light to be reflected back from the tissue of interest. Because light travels too fast to be detected directly, reflected light must be measured indirectly, using a method called low coherence interferometry. Hence, OCT measures the depth of a given structure within the tissue, as well as how much it scatters light. This single measurement is known as an amplitude scan, or A-scan as abbreviation. By scanning the OCT probing beam across the tissue rapidly, multiple OCT A-scans may be obtained and combined to generate a cross-sectional image (B-scan). Volumetric information can be generated by serially capturing multiple B-scans in a region of the retina.

OCT can provide high-resolution, 3-dimensional topographic maps of the retina non-invasively. Given its excellent reproducibility, OCT measurements of retinal thickness are used to quantitatively and qualitatively monitor macular edema to guide the therapeutic intervention of DR (Figure 2). Furthermore, OCT can detect subclinical macular edema, that may otherwise be missed on traditional methods, such as contact and non-contact slit lamp biomicroscopy, indirect ophthalmoscopy and fundus photography. OCT captures structural information within the retina, and it does not provide angiographic information. In the diagnosis and management of diabetic macular edema, OCT is unable to diagnose macular ischemia.

## 9. Optical Coherence Tomography Angiography

Significant improvements in the sensitivity and speed of the OCT imaging platform have led to the development of OCTA, which is capable of providing depth-resolved images of the microvasculature in the retina and choroid at a depth and clarity, coming close to that of histology [68]. OCTA can display the capillary beds at distinct depths, separating the superficial and deep capillary plexuses as well as the choriocapillaris layer, which has increased our understanding of the microvascular changes in DR [69] (Figure 3).

### 9.1. Principle of Optical Coherence Tomography Angiography

OCTA also uses the method of low coherence interferometry, as seen in OCT to gain depth information [68]. The idea of OCTA is to use particle motion within the vessels as intrinsic contrast. It generates the angiograms by comparing motion-related differences between repeated OCT B-scans, taken exactly at the same location. When sequential B-scans of the same location are analyzed for decorrelative signals, stationary structures within the retina will be largely similar (correlated signals), whereas the moving red blood cells that are within the vessels will cause the OCT signal to appear different from one scan to the next (decorrelated signals). These varying differences are translated to blood vessels in the angiograms. Several different approaches were used to analyze the decorrelation between repeated B-scans, using various components of the complex OCT signal [70,71,72], with distinct advantages and disadvantages in each of these approaches. Different vendors have also used different methods for calculating and displaying the OCT angiogram and thus comparability between different machines is not possible [73].

### 9.2. Comparison of Optical Coherence Tomography Angiography and Fluorescein Angiography

Fluorescein angiography requires intravenous dye injection, which can lead to adverse reactions [65]. The risks of adverse reactions are considerably higher in certain patients, such as those with severe kidney diseases. This is because sodium fluorescein is largely removed by the kidneys, hence, patients with renal insufficiency may be at higher risk of nephrotoxicity after fluorescein angiography. Since OCTA is based on flow motion detection, there is no need for contrast dye injections [65]. Therefore, OCTA may serve as an alternative method of angiography that can be safely and more frequently performed to determine the treatment efficacy of DR.

Fluorescein angiography is not able to provide images of the distinct layers of blood vessels, because it is limited to two dimensions and does not provide depth resolution. In contrast, OCTA has the capability of visualizing the distinct retinal vascular layers with high axial resolution [69]. However, OCTA is unable to evaluate the breakdown of the blood-retinal barrier, which is seen as leakage of fluorescein molecules from hyperpermeable pathological vessels in fluorescein angiography. Hyperpermeability in the retinal vascular lesions is an important indication of retinal edema and neovascularization, therefore, fluorescein angiography remains an essential diagnostic modality for DR. However, this absence of the “leakage blur” allows OCTA to generate considerably higher quality images of microvascular structures, even when the vessel wall is leaky [74]. Lesions that have slow flow would not be detected by OCTA such as in subtypes of microaneurysms [74,75,76] and fibrotic neovascularization [77]. Since OCTA relies on contrast between consecutive B-scans, it will detect flow only above a minimum threshold, which is affected by the time between the two sequential OCT B-scans.

### 9.3. Database Search and Results

PubMed literature search was utilized as a source of literature between 1 January 2014 and 31 March 2020. English language articles were retrieved using the keyword search terms, such as optical coherence tomography angiography, diabetes, diabetic retinopathy and diabetic macular edema. The following types of studies were included in the review: cohort, case-control, cross-sectional or intervention trials. We excluded case reports, retrospective, review articles and animal studies. The electronic search yielded a total of 326 citations. After screening, 196 relevant articles were studied, and key findings were extracted.

### 9.4. Optical Coherence Tomography Angiography Visualization of Diabetic Retinopathy Features

Many of the common vascular features of DR, as seen on fluorescein angiography, including microaneurysms, neovascularization, and retinal nonperfusion regions, have been comprehensively studied and described using OCTA [69] (Figure 4).

#### 9.4.1. Microaneurysms

Microaneurysms, which are the first clinically detectable sign of DR, are seen as homogeneous hyperfluorescent punctate spots in fluorescein angiography [78]. In OCTA, microaneurysms can be further delineated into various morphological lesions, such as focally dilated saccular or fusiform capillaries, and are found in the superficial and deep vascular plexuses [76,79]. These microaneurysms, as seen from OCTA, appeared to be similar to those observed in histopathological studies [80]. While OCTA offers a clearer visualization of microaneurysms, the detection rate may be lower in comparison to fluorescein angiography [74,75,76]. This is most likely due to the relative insensitivity of OCTA to the slow blood flow within certain subtypes of microaneurysms [74,75,76].

#### 9.4.2. Neovascularization

On fluorescein angiography, retinal neovascularization is detected on identifying characteristic vessels with excessive leakage in the later phase. However, excessive dye leakage can obscure the vascular details of these abnormal vessels. In OCTA, the contrast depends on erythrocyte movement and the images are acquired over a short time, hence, dye leakages have no impact on the quality of images. As such, the vascular characteristic of neovascularization is displayed with greater clarity in OCTA compared to fluorescein angiography.

Since OCTA can provide information on the various retinal layers, it can help to distinguish between retinal neovascularization, which develops anterior to the retinal vessels and above the inner limiting membrane, and intraretinal microvascular abnormalities, which occur in the same plane as the retinal blood vessels [81,82]. De Carlo and colleagues showed that retinal neovascularization often appeared next to intraretinal microvascular abnormalities [83]. Therefore, OCTA may help to detect subtle neovascularization, which is difficult to differentiate from intraretinal microvascular abnormalities on clinical examination [81,82].

While OCTA is unable to provide information on vascular leakage, morphologic evaluation of neovascularization using OCTA may be able to estimate the activity status of the neovascularization. Ishibazawa and co-workers reported that exuberant vascular proliferation (irregular proliferation of fine new vessels) in OCTA should be considered as a sign of active neovascularization [77]. Hence, quantitative investigation of the extent of retinal neovascularization with OCTA can be used to guide effective therapeutic strategies [84].

#### 9.4.3. Peripheral Retinal Nonperfusion

With fluorescein angiography, nonperfusion regions are seen as dark areas, with loss of capillaries surrounded by larger retinal vessels. OCTA can visualize these corresponding areas of nonperfusion within the superficial vascular plexus and the deep vascular plexus [74,81,85]. Previous qualitative studies in DR have shown that OCTA is capable of delineating retinal capillary nonperfusion with better resolution than fluorescein angiography, providing an improved visualization of capillary dropout and changes in the foveal avascular zone (FAZ) [74,79,82,85]. However, the nonperfusion areas, as seen on OCTA, may either represent capillary occlusion, capillary dropout (complete loss of capillaries) or perfusion deficits (presence of extremely slow flow or absence of flow within the existing retinal capillary) and cannot be differentiated [81]. Changes in vessels visualized on OCTA images do not necessarily indicate structural changes to the blood vessel angioarchitecture and capillary dropout, because the OCTA angiograms depict perfused vessels only. When the blood flow is very slow in diseased eyes, the decorrelation values may be below background noise floor, and therefore remain undetected [81].

Widefield fluorescein angiography revealed that peripheral retinal nonperfusion is a common finding in eyes with DR [86]. These peripheral nonperfusion lesions have been associated with higher risks of DR progression and support the hypothesis that peripheral nonperfusion may be a useful surrogate for and potential predictor of proliferative DR [87,88,89]. Therefore, numerous researchers have explored the use of widefield OCTA to identify peripheral capillary nonperfusion [90,91,92,93]. They reported that widefield OCTA shows comparable diagnostic performance to that of widefield fluorescein angiography for retinal nonperfusion areas [90,91]. (Figure 5) Tan and co-workers further improved the diagnostic performance of widefield OCTA in detecting nonperfusion areas, by removing the influence of larger retinal vessels from capillaries in OCTA scans [92]. Furthermore, widefield OCTA resulted in the higher detection of retinal neovascularization than on clinical examination, which suggests that widefield OCTA could be considered for the purpose of early detection of neovascularization [94,95]. Of note, the widefield fluorescein angiography remains a vital clinical tool in its ability to detect both peripheral retinal nonperfusion and eventual peripheral active neovascularization, which remains difficult to visualize clinically and is less accurately identified with widefield OCTA.

#### 9.4.4. Foveal Avascular Zone Alterations

In the retina, the center of the macula is generally capillary-free; this region is named as the foveal avascular zone (FAZ) and is encircled by a foveal capillaries ring. Alterations to the perfusion of the foveal capillaries greatly affect the patient’s vision [96]. With fluorescein angiography, the FAZ appeared to be larger [97,98] and more irregular [99] in diabetics compared to controls.

The shape and size of FAZ are found to be comparable between OCTA and fluorescein angiography [100]. FAZ metrics can be more easily measured with OCTA than fluorescein angiography, because it is not obscured by fluorescein from dye leakage [85]. Several studies have demonstrated significant quantitative differences in the FAZ in DR compared to normal controls. In diabetics, the FAZ becomes enlarged as a result of loss of integrity of blood vessels [101,102,103]. Furthermore, the outline of FAZ is irregular, due to gaps of the capillary plexuses and the vascular abnormalities are more evident in the deep capillary plexus [101]. Therefore, the FAZ worsens with DR severity, suggesting that the quantitative assessment of the FAZ area may serve as a potential biomarker for macular ischemia in DR [104].

### 9.5. Quantitative Measurements in Diabetic Retinopathy

Several methods to quantify OCTA vascular density outcomes have included perfusion density (or vessel area; calculated as the percentage of the area occupied by vessels; Figure 6B,E), vessel density (or vessel length; calculated as the total length of skeletonized vessels in an area; in mm/mm^2^
Figure 6C,F). These two parameters are widely used in DR studies, as they are commercially available in OCTA devices [104,105,106,107,108,109,110]. Other vascular parameters have also been described, including vessel diameter index (the average vessel caliber), fractal dimension (an index of the branching complexity of the capillary network), intercapillary area, vessel length fraction (total length of vessels), vascular architecture (such as branching angles, tortuosity and fractal dimension), and nonperfusion indexes [104,111,112,113,114]. Apart from static vascular biomarkers, another promising OCTA biomarker is vascular reactivity (the dynamic response of the vessels) [115].

### 9.6. Quantitative Measurements are Correlated with Severity of Diabetic Retinopathy and Visual Acuity

Numerous quantitative OCTA measures of either flow areas (i.e., vessel density and perfusion density) or nonflow area (i.e., nonperfused areas), have agreed closely with the grading of DR based on clinical features and worsening of visual acuity [104,105,106,107,108,109,110]. They reported that the FAZ area increased and retinal vessel density decreased significantly in all plexuses with worsening levels of DR [104,105,106,107,109,110]. Some found that FAZ enlargement with OCTA was associated with worse visual acuity [108,116], whereas others did not observe this relationship [117]. A reduction in retinal vessel density in both the superficial and deep plexuses has been correlated to visual acuity in diabetic patients [108]. Changes in OCTA measures are correlated significantly with disease severity in eyes with DR, thereby suggesting a promising role for OCTA as an objective technique for monitoring disease progression in DR [104,105,106,107,108,109,110].

Of note, vessel density at the deep capillary plexus showed the strongest correlation with DR severity [105,118] and visual acuity [119], emphasizing the important role of the deep capillary plexus in DR. The deep capillary plexus may be more susceptible to ischemic damage, because it may reside in a watershed zone, where the deep layer of the retinal circulation sits next to high oxygen requirements of the outer plexiform layer [120]. The high oxygen requirements of the photoreceptors may be particularly relevant to DR, as Scarinci et al. showed significant relationships between areas of capillary nonperfusion of the deep capillary plexus with photoreceptor disruption [121]. The authors further found a reduction in retinal sensitivity in the corresponding area of capillary nonperfusion in the deep plexus associated with disorganization of the photoreceptors [122]. This would suggest that the reduction in blood flow at the deep capillary plexus may have functional consequences for photoreceptors and visual function in diabetic individuals.

### 9.7. Optical Coherence Tomography Angiography May Reveal Diabetic Retinopathy Before it is Clinically Detectable

Researchers have drawn mixed conclusions in diabetic eyes without retinopathy when using the OCTA-derived measurements. When using FAZ metrics, some reported a small but significant enlargement of the FAZ in diabetic eyes without retinopathy compared to healthy eyes [103,123,124]. However, others reported no significant difference in the FAZ between the two groups [105,109,125,126,127,128,129,130,131,132]. When using retinal vascular density, some showed a significant reduction in vascular density in both the superficial and deep retinal capillary plexuses [124,131,132,133], while others [127,134] reported significant reduction in vascular density in the deep, but not in the superficial plexus in diabetic eyes without retinopathy, compared to healthy eyes. In contrast, other researchers demonstrated no significant differences in either of the superficial or deep capillary plexuses between the two groups [105,126,130].

Other OCTA metrics, such as areas of retinal nonperfusion, drew conflicting results as well. Nesper et al. found larger areas of retinal nonperfusion in eyes of patients with diabetes without DR than in healthy eyes [105]. These results fit well the results of De Carlo et al. [123], who showed that retinal nonperfusion occurred more often in eyes of patients with DM without DR than in healthy eyes. However, a recent publication by Dai et al. did not observe any significant differences in the nonperfusion area between the two groups [130].

### 9.8. Vascular Changes in the Choriocapillaris in Diabetic Retinopathy

Even though DR is generally considered to be a disease of the retinal vasculature, histopathological and dye-based angiographic studies have shown choroidal changes to be associated with diabetes [135,136,137,138,139,140,141]. These alterations include vascular dropout, areas of vascular nonperfusion, and choroidal neovascularization, occurring predominantly in the choriocapillaris and termed as diabetic choroidopathy [135,136,137,138,139,140,141]. A functional study using laser Doppler flowmetry has shown a significant reduction of subfoveal choroidal blood flow in diabetic individuals, especially in those with macular edema [142]. The dropout of the choriocapillaris as seen in the histological findings could increase vascular resistance [135,136,137,138,139,140,141], thereby causing the reduction in blood flow in the choriocapillaris [142].

Although the concept of diabetic choroidopathy is well-established, the role of the choriocapillaris in DR remains poorly understood [143]. This may be partially due to the location and structure of the choriocapillaris, making it difficult to observe with both fluorescein and indocyanine green angiography [144]. In contrast, OCTA is depth resolvable, allowing the choriocapillaris to be viewed separately from the retinal vasculature. Furthermore, swept-source OCTA technology, which uses a longer wavelength allowing for deeper signal penetration with a mitigated sensitivity roll-off, offers better visualization of the choriocapillaris [145].

Studies have reported vascular abnormalities in the choriocapillaris in diabetic eyes compared to healthy eyes [105,146]. Regions of choriocapillaris flow impairment in patients with DR were observed and these abnormalities were correlated with DR severity [105,146,147] (Figure 7). Dodo and colleagues also demonstrated that impaired flow in the choriocapillaris was associated with photoreceptor damage, suggesting that the impairment of choriocapillaris might result in a deficiency in oxygen or nutrients, thereby contributing to photoreceptor injury [147]. Choriocapillaris flow impairment was also evident in diabetic patients who were without retinopathy using either swept-source [130,146] or spectral-domain [105,131] OCTA systems. Consequently, OCTA may prove to be an invaluable tool for studying the choriocapillaris and may provide new insights into the role of the choriocapillaris in the pathophysiology of diabetic eye disease.

### 9.9. Vascular Changes in the Optic Nerve Head in Diabetic Retinopathy

Only a few studies have used OCTA to investigate the blood flow of the optic nerve head in diabetics [148,149,150,151,152,153]. A significant decrease in the peripapillary vessel density was found in those with DR compared to healthy controls, which indicates that the peripapillary region may be particularly susceptible to damage from diabetes [150,152,153]. Whether this is a consequence of retinal nerve fiber layer thinning and the associated decrease in metabolic demand is unknown. However, more studies are required to examine the consistency of peripapillary retinal nerve fiber layer measurements using various OCTA systems [154], while recognizing potential issues with segmentation and analysis of OCTA scans of optic nerves with pathologies such as disc hemorrhages or edema [155]. The reduction of the optic nerve head vessel density correlated with the increasing severity of DR, which may explain the development of disc neovascularization [150,152]. However, Rodrigues et al. did not observe any correlation between peripapillary vascular network with DR severity when he excluded the larger blood vessels from the quantification of peripapillary capillary networks [153]. The reduction of the optic nerve head vessel density was correlated with worsening levels of visual acuity [151]. Notably, some researchers found a significant decrease in the peripapillary vessel density in diabetics, even before the appearance of clinically apparent signs of DR when compared to normal controls [148,149,152,153], whereas others did not observe such differences [150].

### 9.10. Confounding Factors in Optical Coherence Tomography Angiography Studies

In studying the vascular changes of DR with OCTA, it is essential to consider the effects of confounding factors. OCTA metrics can be confounded by the effects of axial length and individuals’ systemic vascular risks. Axial length can affect the lateral magnification of OCTA images and consequently can affect the accuracy of quantitative OCTA metrics [156,157]. This is mainly dependent on how much FAZ one includes in the analytical regions (Figure 8). In eyes with shorter axial length, the OCT scan area is smaller than that of a myopic eye. A smaller scan area would mean that the FAZ occupies a larger portion of the scan area, resulting in an erroneously lower vessel density [158]. This is particularly crucial for the vessel density of the deep capillary plexus, because the FAZ in the deep capillary plexus is considerably larger than in the superficial vascular plexus [159]. Two potential solutions have been proposed to overcome the effect of OCT magnification. The first solution would be to rescale the OCT, which will require axial length to be measured. Another solution would be to assess the irregularity of the FAZ without the need of rescaling. Both of these newly proposed measures, which have been termed as the acircularity index and axis ratio respectively, have been found to be correlated to the current ETDRS staging system for DR [108,128,160].

The quantification of OCTA metrics can also be confounded by interindividual variation, age and systemic vascular risks. For example, OCTA metrics such as retinal capillary density, choriocapillaris flow impairment, FAZ dimensions are known to vary even in normal persons [161,162,163,164]. Due to a lack of normative values, it remains difficult to decide when a diabetic patient’s FAZ size should be considered as abnormal. Overall, the dimensions of FAZ have numerous limitations to serve as biomarkers of DR severity, given its physiologic variability and segmentation/measurement limitations [161,162,165,166,167].

Furthermore, patients’ systemic disease status and medication use may cause a decrease in blood flow and alter the entire OCTA image. Some early studies have reported that systemic blood pressure levels [168,169], and diabetic control [170] are associated with OCTA metrics. Therefore, future studies will need to account for multiple confounders that may also affect vascular changes, such as how long it has been since the patient was diagnosed with diabetes, and other systemic vascular risks, to better understand the relationship of OCTA metrics in DR with systemic control.

## 10. Limitations of Optical Coherence Tomography Angiography

The clinical applicability of OCTA is limited by several factors. Since OCTA is based on motion detection, it is particularly sensitive to the patient’s eye movement (Figure 9). The need for good patient fixation can be challenging in diabetic patients with macular involvement. Incorporating eye tracking during OCTA scanning can mitigate the eye movement [171,172]. However, the use of an eye tracking system may lead to longer image acquisition time. Alternatively, motion correction techniques can minimize motion artifact as well [173].

The limited field of view with current OCTA results is another limitation of OCTA in the detection of peripheral vascular changes, such as peripheral retinal nonperfusion and neovascularization. Current OCTA can image areas of 12×12 and 15×9, however these remain far from comparable to ultra-widefield angiography. To increase the field of view of OCTA, one can montage multiple images together [174]. In addition, fast prototype system with angles up to 100 degrees have been realized [175].

Another limitation of OCTA is in its inability to quantify blood flow. Current OCTA has a sensitivity threshold and therefore it will detect flow only above a minimum threshold, otherwise known as the slowest detectable flow [68]. Moreover, if flow is fast, the OCTA image saturates, the fastest distinguishable flow is also limited. An advantage of a higher speed system is that more repeated OCT volumetric data can be captured at each cross-section, thereby reducing the interscan time, which will then allow faster flow to be distinguished [68]. In a fast system, a low-flow vessel may be undetected when using the first and second OCT B-scans; the image may be processed using the first and third B-scans to increase the interscan time, thereby allowing slow flow to be detected [176]. Therefore, customizing the slowest detectable flow and fastest distinguishable flow will allow a better investigation of changes in pathological conditions [176].

Artefacts can affect the accurate quantification of vascular changes. As seen from Figure 5A, these yellow areas were segmented out in the normal eye, because they were areas of low intensity and not of low perfusion. One such artefact is known as vignetting, which creates an uneven illumination of the image, where the center of the image appears brighter than that of the periphery. Images obtained from subjects with media opacities, such as cataracts and poorly dilated eyes, tend to suffer from optical vignetting. Diabetic patients often have pupils which dilate poorly to mydriatic agents [177]. The accurate identification of low contrast regions from pathological capillary dropout is crucial for detecting DR eyes with poor perfusion. Pretto et al. used spatial variance filter and OCT contrast thresholding to differentiate the artifacts from real capillary dropout [178]. There is an intense area of research on the development of algorithms that can minimize artefacts to improve the clinical utility of the OCTA.

The accuracy of segmentation algorithms in the identification of retinal layers, especially in eyes with pathological abnormalities, imposes the limitation in OCTA data interpretation. The occurrence of segmentation error was 11% in the non-proliferative DR category and increased to 50% in the proliferative DR category [179]. Furthermore, high rates of segmentation error, particularly in eyes with diabetic macular edema, that may affect vascular density measurements, were reported [180]. Correction of these segmentation errors can be made manually with the built-in software. However, such correction would be tedious especially when several hundreds of B-scan images would need to be manually corrected for each volumetric scan. Future advancements will likely provide a more efficient tool that can automatically segment irregular layers.

Another limitation of existing OCTA devices is the derivation of measures related to the deep capillary plexus. Studies have suggested that impairment in the deep capillary plexus may play a critical role in vision loss in DR [105,118,119]. However, OCTA images of the deep capillary plexus are prone to projection artefacts, which are a result of the high scattering dynamics of blood within the vessels of the superficial capillary plexus, and can interfere with the interpretation of retinal angiographic results. Despite having the projection removal function in the built-in software [181], residual project artefacts still exist (Figure 10). There is active research on the development of post-processing algorithms that can reduce projection artefacts to provide better visualization of OCTA images [181,182,183].

Different quantitative OCTA metrics have been introduced, such as vessel density, area of nonperfusion and choriocapillaris flow voids. However, wide discordance exists in the way these OCTA indices are calculated. A detailed overview of how different quantitative measures are derived from OCTA is beyond the scope of this review, but the reader is referred to a recent review article [184]. Briefly, although measurements by the same devices using the same processing algorithm may be reliable and reproducible [185,186,187,188], significant variability exists between different devices [189] and image processing methods [186]. Therefore, large population studies of multi-ethnic individuals will need to be collected for each OCTA device, to provide normative data of vascular measures as reference ranges. Additionally, there is a need to develop a better nomenclature on how various OCTA parameters should be defined and derived, thereby improving the clarity of concepts regarding the use of OCTA for DR imaging.

## 11. Future Studies

The majority of the studies mentioned here are cross-sectional in nature, except for one study, where Sun et al. demonstrated the predictive value of OCTA measures for progression of DR [190]. Moreover, most of the studies stated here have a modest number of patients with a sight-threatening form of DR category, and have not reported traditional risk factors such as HbA1c, duration of diabetes, blood pressure levels, diabetic medication history, and the presence of co-morbid diseases that may confound the vascular changes [191]. Another potential use of the OCTA is the real time imaging of the retinal vasculature during exposure to hyperoxia or hypercapnia hypercapnic conditions or flicker light stimulation [115,192]. The measurement of vascular reactivity with OCTA may potentially be a clinical feasible biomarker for the early diagnosis of DR [33]. Another potential area of future development is the use of OCTA to detect iris neovascularization development in proliferative DR [193,194]. As complications from iris neovascularization development can be sight-threatening, prompt intervention is required and OCTA may be a useful modality to detect response to anti-VEGF treatment [195]. The incorporation of artificial intelligence/ deep learning on OCTA images may have the potential to play a significant role in the diagnosis and management of DR in the near future [196]. One recent study used a deep learning approach and effectively detected the non-perfusion area in OCTA images of DR [197]. Further studies employing longitudinal multicenter design, greater patient numbers with sight-threatening forms of DR, as well as the consideration of relevant confounders, will improve our understanding of the relationship between the benefit of OCTA biomarkers in the detection and progression of DR.

## 12. Conclusions

The future of OCTA technology for the imaging of DR looks promising. The qualitative visualization of vascular changes in varying stages of DR has improved our understanding of the pathophysiology of DR. Quantitative OCTA measurements can objectively grade progressive vascular changes at the retinal, choroidal and optic nerve, highlighting its potential to be used as an objective tool for monitoring of DR progression. Of interest is the capability of OCTA to detect DR even before it is clinically detectable, highlighting its possible use as an early screening tool for DR. Novel widefield OCTA technology can provide a wider view of the peripheral retina and can improve the detection of peripheral retinal nonperfusion and neovascularization.

## Figures and Tables

**Figure 1 jcm-09-01723-f001:**
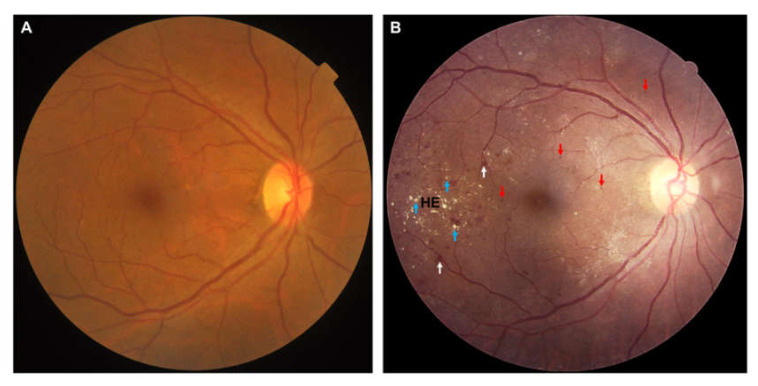
(**A**) Color fundus photograph of the right eye of a 58-year-old woman, without any chronic systemic diseases or eye diseases. (**B**) Color fundus photograph of the right eye of a 45-year-old man with signs of moderate non-proliferative diabetic retinopathy. Notably, features include microaneurysms (red arrows), dot- and blot- hemorrhages (white arrows) and hard exudates (HE, blue arrows).

**Figure 2 jcm-09-01723-f002:**
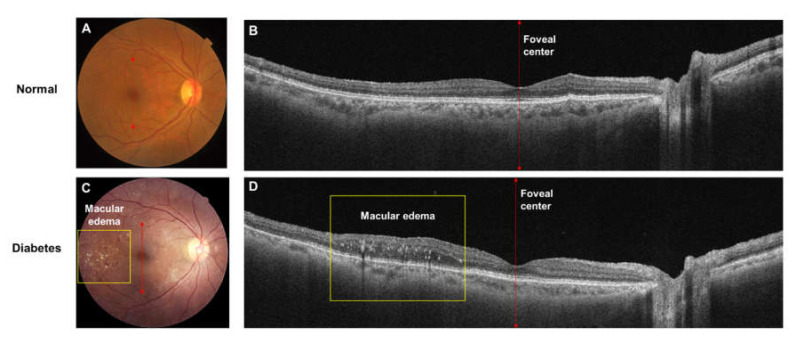
(Top row; A, B) Images of the right eye of a 58-year-old woman, without any chronic systemic diseases or eye diseases. (**A**) Color fundus photograph. (**B**) widefield optical coherence tomography (OCT) of a horizontal scan through the center of the fovea (red line) revealing a normal retina. (Bottom row, **C**,**D**) Images of the right eye of a 45-year-old man with signs of moderate non-proliferative diabetic retinopathy. (**C**) Notably, features include microaneurysms, dot- and blot- hemorrhages and hard exudates (yellow box). (**D**) Wide field OCT of a horizontal scan through the center of the fovea (red line) reveals marked thickening of the retina at the temporal quadrant of the retina (yellow box).

**Figure 3 jcm-09-01723-f003:**
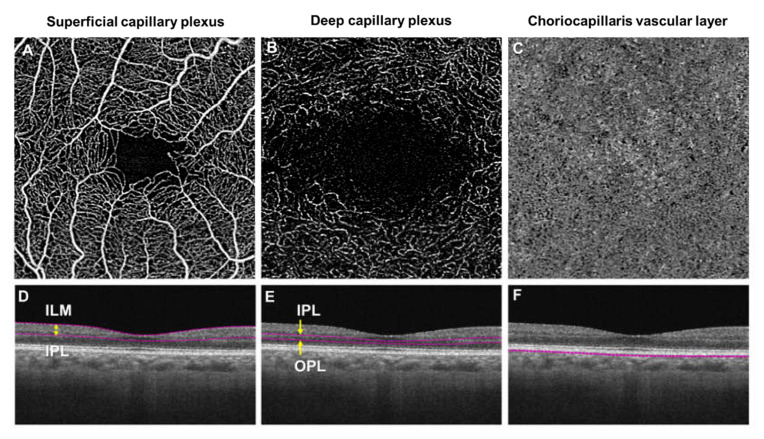
En-face optical coherence tomography angiography (OCTA; 3 × 3 mm area) images (Top row, **A**–**C**) and horizontal B-scan images of their layer-segmentation (Bottom row, D–F) of a healthy control individual. (**A**) OCT angiogram of a superficial vascular plexus centered on the macula. (**B**) The OCT angiogram of a deep capillary plexus centered on the macula. (**C**) An OCT angiogram of a choriocapillaris vascular layer centered on the macular. (**D**) The en-face image of the superficial plexus was segmented from the internal limiting membrane (ILM) to the inner plexiform layer (IPL). (**E**) The image of the deep plexus was segmented from the IPL to the outer plexiform layer (OPL). (**F**) The OCT angiogram of the choriocapillaris layer was segmented within a thin 10 µm thick slab (31–40 µm) below the retinal pigmented epithelium. The foveal avascular zone (FAZ) is visibly larger in the deep plexus (**B**) than superficial plexus (**A**). (**B**,**C**) were created after removal of projection artifacts.

**Figure 4 jcm-09-01723-f004:**
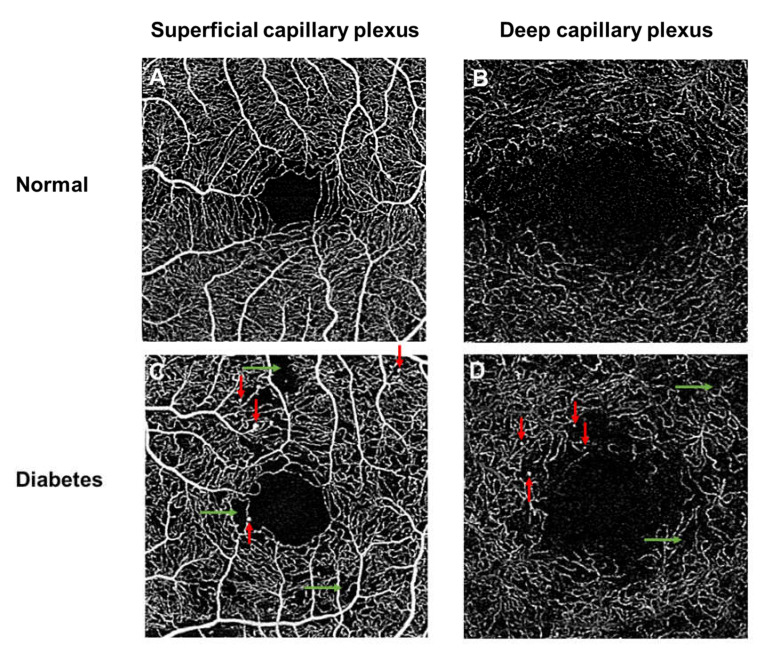
Optical coherence tomography angiography (OCTA; 3 × 3 mm area) of a healthy control individual (Top panel; **A**,**B**), showing an extensive network of capillaries of the superficial vascular plexus, where the foveal avascular zone is surrounded by the foveal capillary network. OCTA images of a patient with diabetes (Bottom panel; **C**,**D**), showing vascular abnormalities in both superficial and deep plexus layers, such as microaneurysms (red arrows), capillary nonperfusion (green arrows). Note the enlarged foveal avascular zone (FAZ). (**B**,**D**) were created after removal of projection artifacts.

**Figure 5 jcm-09-01723-f005:**
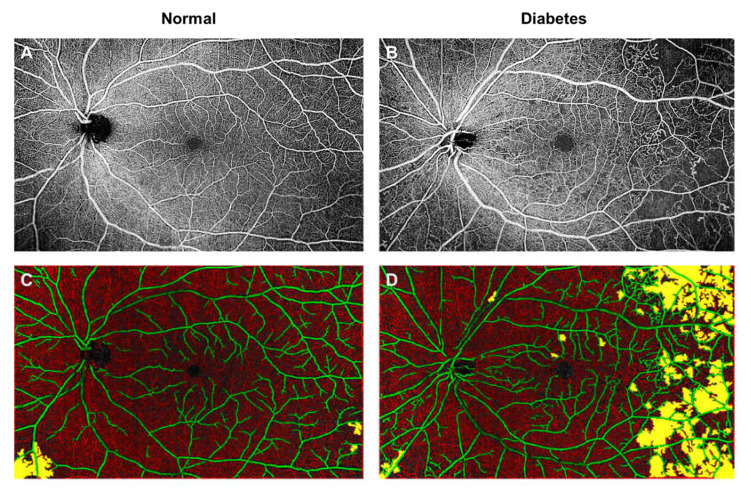
Widefield optical coherence tomography angiography (OCTA; 15 × 9 mm area; **A**,**B**) and color-coded maps indicating regions of low or nonperfusion (**C**,**D**; labelled as yellow) of a superficial vascular plexus of a healthy control individual (Left panel; **A**–**C**) and a diabetic (Right panel; **B**–**D**). There are increased regions of retinal nonperfusion, particularly in the temporal regions in a diabetic eye (D; labelled as yellow). It is important to note that in the normal subject, sporadic yellow areas can be seen in the peripheral region (**C**; labelled as yellow). These yellow areas were segmented out, because they were areas of low intensity and not of low perfusion.

**Figure 6 jcm-09-01723-f006:**
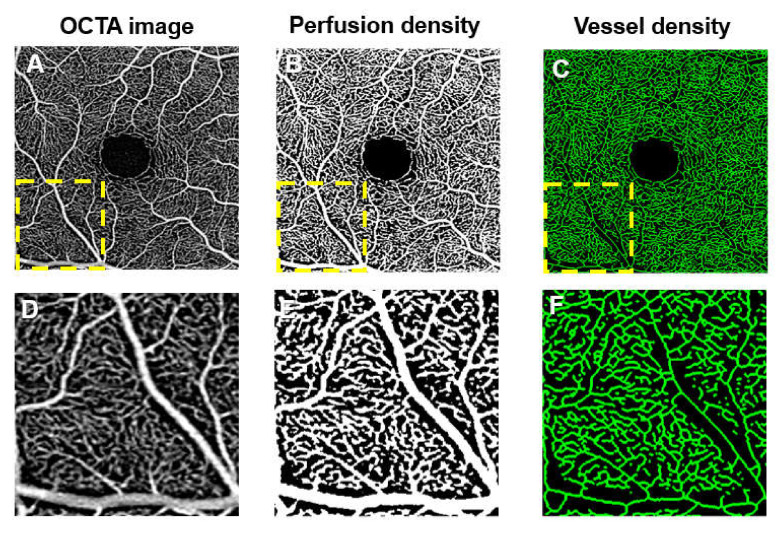
Optical coherence tomography angiography (OCTA) of the superficial capillary plexus (**A**,**D**) and post-processed images (**B**,**C**,**E**,**F**). Corresponding enlarged views of the images are shown in (**D**–**F**). (**A**) OCTA image is first extracted from the machine. (**B**,**E**) The image is then binarized to obtain the perfusion density. (**C**,**F**) To obtain the vessel density, the binarized image is further skeletonized. In perfusion density, large vessels count more towards the perfusion density measurements, while in vessel density, large vessels and small capillaries carry the same weight. This difference can lead to discordances in OCTA vascular density outcomes.

**Figure 7 jcm-09-01723-f007:**
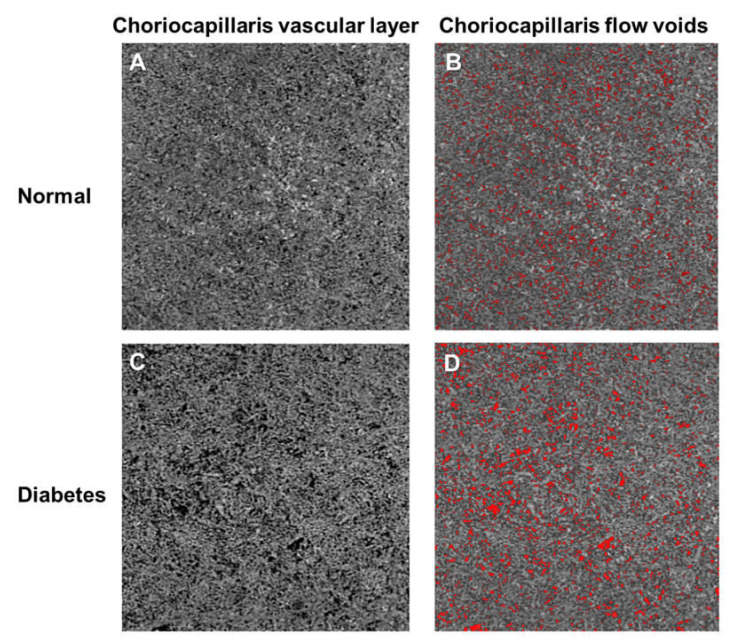
Optical coherence tomography angiography (OCTA; 3 × 3 mm area) and color-coded maps indicating regions of flow voids (**B**,**D**; labelled as red) of a choriocapillaris vascular layer of a healthy control individual (Top panel; **A**,**B**) and a diabetic (Bottom panel; **C**,**D**), showing the presence of more and larger sized choriocapillaris flow voids in a diabetic eye (**D**; labelled as red). The presence of flow voids can be seen as areas of dark regions in the angiogram (**A**,**C**) and are labelled as red in the color-coded map (**B**,**D**). Figures were created after removal of projection artifacts.

**Figure 8 jcm-09-01723-f008:**
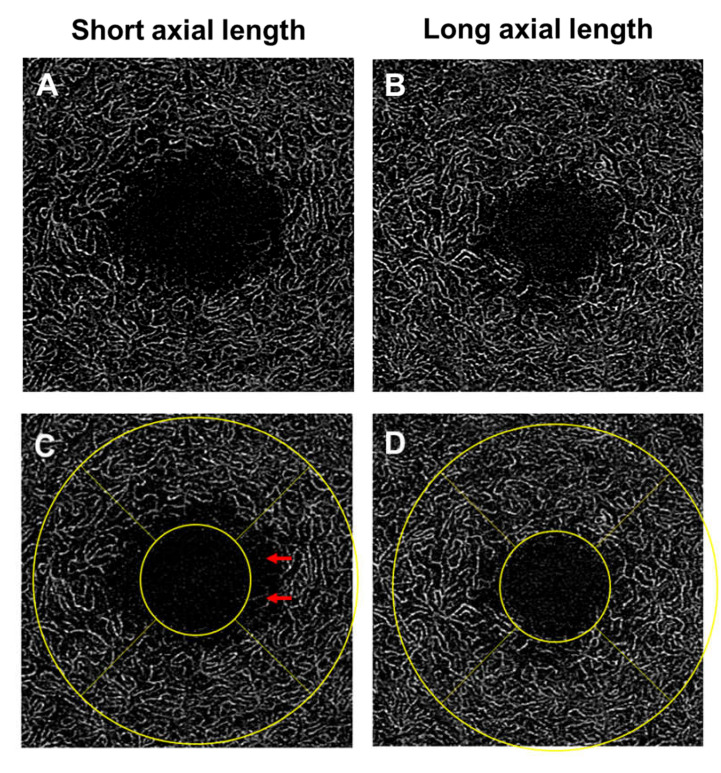
Optical coherence tomography angiography (OCTA; 3 × 3 mm area) of the deep vascular plexus. A hyperopic eye with spherical equivalent of +1.00 DS and an axial length of 23.23 mm (**A**,**C**) and a highly myopic eye with spherical equivalent of −10.25 DS and an axial length of 28.29 mm (**B**,**D**). The analytical area is defined by two concentric circles (1- and 3-mm nominal diameters) and is centered on the foveal avascular zone (FAZ). In eyes with hyperopia (shorter axial length), the OCT scan area is smaller than that of a myope. A smaller scan area would mean that the FAZ occupies a larger portion of the analytical area, resulting in a lower vessel density (red arrows). Conversely, a myopic eye (longer axial length) would have a larger scan area, ultimately an erroneously higher vessel density. Figures were created after removal of projection artifacts.

**Figure 9 jcm-09-01723-f009:**
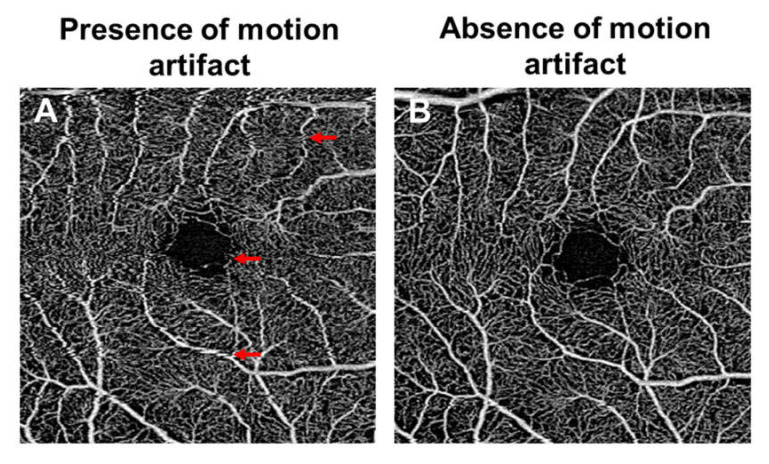
Optical coherence tomography angiography (OCTA; 3 × 3 mm area) of the superficial vascular plexus, showing the presence of motion artifact in the first scan (**A**) and the absence of motion artifact when a second scan was taken (**B**). Eye motion leads to discontinuities in the OCTA image (red arrows) and potentially affects the quantitative data.

**Figure 10 jcm-09-01723-f010:**
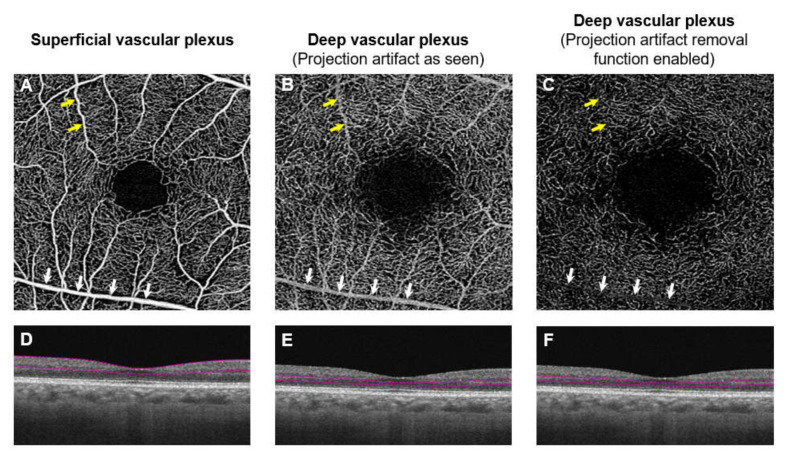
Angiographic images and projection artifacts. (**A**–**C**) are different en face capillary plexus and (**D**–**F**) are their corresponding B-scan images of the same eye. The superficial capillary plexus is shown in A and deep capillary plexus is shown in (**B**,**C**). When the segmentation section is lowered to include the region around the inner nuclear layer and outer plexiform layer, the deep capillary plexus should be seen. However, the vascular pattern seems similar to the superficial capillary plexus. Note the superficial vessels seen in the angiographic image (yellow and white arrows; (**A**,**B**). The superficial vessels are seen because of a projection image. When the projection removal algorithm is enabled, it removes most of the projection image from the superficial capillary plexus. Although smaller superficial vessels are no longer visible (yellow arrows); (**C**) larger superficial vessels still persist (white arrows; **C**).

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
