# Peer review of "Optical Coherence Tomography Angiography in Diabetes and Diabetic Retinopathy"

_jcm, 2020, doi:10.3390/jcm9061723_

Round 1
Reviewer 1 Report
The review submitted by Chua et al., entitled Optical Coherence Tomography Angiography in Diabetes and diabetic Retinopathy, summarizes current knowledge on the use and benefits of OCTA as an alternative method of angiography to detect DR.
After minor revision, the paper can be accepted.
Comments:
- Given that the manuscript is a review on the clinical applicability of OCTA on DR, the Pathophysiology and Stages of Diabetic Retinopathy section should contain more information on the complex mechanism underlying DR.
Moreover, in the Ocular Vascular Changes in Diabetes section, the authors should better explain the role of hypoxia and angiogenesis as key drivers of blood-retinal barrier breakdown. Please quote some studies, e.g. Journal of cellular physiology 233 (2), 1120-1128; Surv Ophthalmol 2009 54:1-32.
- In Fluorescein Angiography section, the last sentence from “Despite the clinical utility..” should be re-phrased.
Reviewer 2 Report
The authors present an interesting and comprehensive review on the usefulness of OCT-A in patients with diabetes.
The paper can be slightly shortened because it should be focused on OCT-A. It will reduce the number of self-citations.
The authors list the limitations of this imaging method. However, obtaining a clear and useful deep capillary plexus (DCP) is even more difficult then Authors seem to suggest. In many patients, the superficial capillary plexus is clearly visible, while DCP imaging cannot provide reliable data. Unfortunately, DCP is much more important in diabetic patients.
There are several different methods for obtaining OCT-A data from the retina, although not all are available on the market. However, they can be mentioned.
Despite these comments, the article is well written and gathers important information.
Reviewer 3 Report
The authors submitted a well written review on the applications of OCTA in diagnosis, classification and management of diabetic retinopathy.
Although comprehensive, a few topics worth editing and further references are included below:
in topic 6. Ocular Vascular Changes in Diabetes
- consider including references that add to the discussion on potential ways of characterising very early vascular changes in DR and signs of neurovascular coupling dysregulation, such as PMID: 31273312, PMID: 32232345, PMID: 31173077, and PMID: 31249500
In point 9.2
- worth mentioning as an OCTA limitation not only the lower threshold for detection in cases of slow blood flow, but also the issues associated with very high blood flows. And cite accordingly (e.g. Spaide’s OCTA review)
In point 9.4.3
- the authors fairly mention the important role of wide-field FA to detect retinal nonperfusion. and mention then that widefield OCTA might have comparable diagnostic accuracy. However, I would add a sentence on the important role of wide-field FA to detect BOTH peripheral retinal non perfusion and eventual peripheral active neovascularization (difficult to see clinically and not possible to accurately identify with widefield OCTA)
point 9.4.4
- FAZ has numerous limitations to serve as a biomarker of DR severity in clinical practice, given its physiologic variability and segmentation/measurement limitations (eg PMID: 31047396, PMID: 31360604, PMID: 31792852, PMID: 30280005, PMID: 28616362).
- Vessel density (or vascular reactivity - eg PMID: 31249500) are much more promising in this sense.
point 9.6
- again, worth commenting on the less valid option of using FAZ metrics for this purpose.
- suggest citing other recent studies (eg. 31273312) and commenting on the potential measurement bias associated with less recent studies (eg lack of Projection artefact removal software) - to explain the lack of total agreement between different groups - with a suggestion of the ideal way of doing it?
point 7 - multiple limitations in assessing te choriocapillaris with OCTA… also related with the physiologic variability… which might be worth mentioning.
point 9 - well-done, this covers some of the previous comments/suggestions, please consider adding appropriate referencing.
point 11 - suggest including the potential role of OCTA to measure retinal vascular reactivity, which would overcome the baseline differences among individuals by measuring a response and not a single static vessel density value.
good job overall - congratulations to the authors.
